# Heterogeneity of Amino Acid Profiles of Proneural and Mesenchymal Brain-Tumor Initiating Cells

**DOI:** 10.3390/ijms24043199

**Published:** 2023-02-06

**Authors:** Corinna Seliger, Lisa Rauer, Anne-Louise Wüster, Sylvia Moeckel, Verena Leidgens, Birgit Jachnik, Laura-Marie Ammer, Simon Heckscher, Katja Dettmer, Markus J. Riemenschneider, Peter J. Oefner, Martin Proescholdt, Arabel Vollmann-Zwerenz, Peter Hau

**Affiliations:** 1Department of Neurology, University Hospital Heidelberg, 69120 Heidelberg, Germany; 2Department of Neurology and Wilhelm Sander-NeuroOncology Unit, University Hospital Regensburg, 93053 Regensburg, Germany; 3Department of Psychosomatic Medicine and Psychotherapy, University Medical Center Freiburg, 79104 Freiburg, Germany; 4Institute of Functional Genomics, University of Regensburg, 93053 Regensburg, Germany; 5Department of Neuropathology, University Hospital Regensburg, 93053 Regensburg, Germany; 6Department of Neurosurgery, University Hospital Regensburg, 93053 Regensburg, Germany

**Keywords:** glioma, metabolism, metformin, proneural and mesenchymal brain-tumor-initiating cells

## Abstract

Glioblastomas are highly malignant brain tumors that derive from brain-tumor-initiating cells (BTICs) and can be subdivided into several molecular subtypes. Metformin is an antidiabetic drug currently under investigation as a potential antineoplastic agent. The effects of metformin on glucose metabolism have been extensively studied, but there are only few data on amino acid metabolism. We investigated the basic amino acid profiles of proneural and mesenchymal BTICs to explore a potential distinct utilization and biosynthesis in these subgroups. We further measured extracellular amino acid concentrations of different BTICs at baseline and after treatment with metformin. Effects of metformin on apoptosis and autophagy were determined using Western Blot, annexin V/7-AAD FACS-analyses and a vector containing the human LC3B gene fused to green fluorescent protein. The effects of metformin on BTICs were challenged in an orthotopic BTIC model. The investigated proneural BTICs showed increased activity of the serine and glycine pathway, whereas mesenchymal BTICs in our study preferably metabolized aspartate and glutamate. Metformin treatment led to increased autophagy and strong inhibition of carbon flux from glucose to amino acids in all subtypes. However, oral treatment with metformin at tolerable doses did not significantly inhibit tumor growth in vivo. In conclusion, we found distinct amino acid profiles of proneural and mesenchymal BTICs, and inhibitory effects of metformin on BTICs in vitro. However, further studies are warranted to better understand potential resistance mechanisms against metformin in vivo.

## 1. Introduction

Glioblastoma (GB) is one of the most aggressive types of human cancer [1]. Despite enormous research efforts over the last decades, median overall survival still ranges between about 15 and 20 months for patients treated with radio- and chemotherapy following brain tumor surgery with or without tumor treating fields [2,3,4]. Brain-tumor initiating cells (BTICs) are supposed to drive tumor cell malignancy and to account for the observed resistance to standard therapies.

One of the pathophysiological features of GB is aberrant tumor metabolism [5]. Whereas glucose metabolism has been extensively explored [6,7], there is still limited data on amino acid metabolism. In our previous work, we were able to demonstrate the metabolic preferences of distinct metabolic subtypes of glioma. We found mesenchymal BTICs to be more glycolytic, whereas proneural BTICs catabolize glucose more towards the pentose phosphate pathway [7]. GBs also use amino acids such as glutamine to meet their energetic needs, especially when glucose availability is limited [8]. Furthermore, GBs are heterogeneous brain tumors and molecular subtypes of GB, namely proneural and mesenchymal GBs, have different abilities to use and build up nutrients [9,10,11].

Metformin is an oral hypoglycemic agent that is under investigation as a potential antineoplastic agent [12]. Metformin has been found to inhibit proliferation and migration of BTICs, especially of the proneural subtype, by altering glucose metabolism in a subtype-specific way [7]. Whereas glucose metabolism and respiration under treatment with metformin have been extensively studied in glioma [7,13], though with partly contradictory results, little is known about the effects of metformin on amino acid metabolism. However, a better understanding of the holistic landscape of glucose and amino acid metabolism in GB is crucial for the development of novel therapeutic strategies and may help to understand resistance to current treatment options, with potential translation into clinical applications.

To that end, we explored the amino acid preferences of GB subtypes. GB cells were explored under normal cell culture conditions and after the application of metabolic stress using metformin as an inhibitor of oxidative phosphorylation (OXPHOS) [14], with the aim to not only investigate the effects on amino acid metabolism, but also on functional outcomes such as apoptosis and autophagy.

## 2. Results

### 2.1. ^13^C-Glucose Tracing in Proneural and Mesenchymal BTICs

In a previous study [7], we described the metabolic preferences of proneural and mesenchymal BTICs and their response to the metabolic drug metformin. For that purpose, we performed glucose flux analyses, which demonstrated that mesenchymal BTICs are more glycolytic while proneural BTICs metabolize glucose more towards the pentose phosphate pathway. As that study was merely based on glucose metabolism and as it is known that glucose and amino acid metabolism are closely interrelated, we aimed at investigating amino acid metabolism more closely.

We used [^13^C6]glucose as a tracer substrate to detect differences in the utilization of glucose-derived carbons in amino acid biosynthesis in the proneural BTIC-18 and mesenchymal BTIC-11 and BTIC-13 (Figure 1). Characteristics of BTICs have been described elsewhere [7]. Based on the published data, we expected that amino acid biosynthesis may vary between different BTICs of proneural and mesenchymal origin [15].

Indeed, serine and glycine tracing revealed that proneural BTIC-18 redirected far more glucose-derived carbons into the biosynthesis of serine and glycine than mesenchymal BTIC-11 and BTIC-13. On the other hand, BTIC-11 and BTIC-13 redirected more glucose into the biosynthesis of aspartate and glutamate.

Treatment with metformin led to a dose-dependent inhibition of amino acid fluxes in nearly all of the investigated cell lines. In most lines, inhibitory effects were observed starting at a dose of 1 mM metformin. Whereas high-dose metformin led to a strong reduction in the amino acid fluxes of all BTICs, ^13^C flux from glucose into alanine was only moderately changed in mesenchymal BTIC-11 and BTIC-13 and even increased in proneural BTIC-18. Furthermore, in BTIC-18, ^13^C flux into serine and glycine was reduced only moderately even after high-dose treatment with 10 mM metformin.

### 2.2. Extracellular Amino Acid Concentrations

To further substantiate our hypothesis that different BTICs of proneural and mesenchymal origin use distinct amino acid pathways, we first explored amino acid metabolism of two proneural BTICs (BTIC-8 and BTIC-18) and two mesenchymal BTICs (BTIC-11 and BTIC-13) and their differentiated tumor cell (TC) counterparts. Cells were maintained in a cell culture medium supplemented with 2 mM L-glutamine and 1 mM glucose, and non-essential amino acids (1% *v*/*v*) were supplemented into cell culture.

Extracellular amino acid levels were measured in cell culture supernatants of either untreated or metformin-treated cells. There was significant heterogeneity among the BTICs and TCs investigated. The heatmap shows amino acid concentrations of essential (threonine, valine, methionine, tryptophan, histidine, leucine, isoleucine, phenylalanine) and non- or conditionally non-essential amino acids (arginine, glutamine, serine, asparagine, glycine, alanine, proline, aspartate, glutamate, ornithine, tyrosine, and cysteine (CC)) (Figure 2).

Relevant differences in amino acid concentrations were observed for several amino acids under different treatment conditions, among those serine, glutamine, aspartate, and glutamate. In addition, when comparing amino acid levels in supernatants with or without tumor cells, we observed that some amino acids were consumed from the cell culture media, whereas other amino acids were produced by the cells (Figure 3).

Extracellular serine levels were higher (all BTICs and TC-8 and TC-18) or similar (TC-11 and TC-13) to the cell culture medium incubated in the absence of cells, indicating that most cells secreted more serine than they potentially took up. Interestingly, all cell lines hardly took up any glutamine. In contrast, almost all proneural cell lines (BTIC-8, BTIC-18, and TC-8) consumed (nearly) all the extracellular glutamate and aspartate available. Aspartate and glutamate are metabolites used to fuel the tricarboxylic acid cycle (TCA).

Only TC-11 excreted glutamate and generated significant amounts of glutamate out of glutamine, while the remaining BTIC and TC cell lines consumed far less aspartate and glutamate than the proneural cell lines, except for TC-18.

Besides comparing amino acid concentrations in the cell culture media of untreated cells, we also compared the amino acid concentrations of cells treated with increasing doses of metformin, namely 0.01 mM tid, 1 mM, and 10 mM metformin. Addition of metformin, regardless of the dose applied, had no impact on uptake. However, 10 mM metformin increased serine secretion in BTIC-11 and decreased glutamate and aspartate release in TC-11.

### 2.3. mRNA Microarrays of Metabolic Hallmarks from Proneural and Mesenchymal BTICs

To explore if the different amino acid preferences of our selected proneural and mesenchymal BTICs may be explained by genetic differences between the proneural and mesenchymal subtype, we performed a gene set enrichment analysis that was based on the mRNA expression of 36 published BTICs, including those used for the present study [16]. BTICs were divided into proneural and mesenchymal BTICs [9] and investigated in hallmark analyses (Figure 4). In our published work, we found that the hallmark glycolysis was significantly distinct in proneural vs. mesenchymal cell lines [7]. Now, we investigated the other hallmarks that are related to tumor metabolism, namely the hallmarks lipid metabolic process, hypoxia, and oxido-reductase activity. In all three hallmarks, we found significant differences between proneural and mesenchymal BTICs (Figure 4). Significantly changed genes of the hallmarks are depicted in Appendix A.

We next screened the gene lists for genes related to amino acid metabolism. Interestingly, we found glycine dehydrogenase, the key enzyme of the glycine cleavage system [17], to be most enriched in proneural BTICs in the hallmark oxidoreductase activity. In addition, aldehyde dehydrogenase 1A3 and 1A1, which are detoxification enzymes as they convert aldehydes to carboxylic acids [18], were enriched in mesenchymal BTICs in the hallmark oxidoreductase activity and lipid metabolic process.

### 2.4. Effects of Metformin on Apoptosis and Autophagy of BTICs

Next, we explored whether the reduced flux of glucose-derived carbons into various amino acids led to increased autophagy or cell death of BTICs and TCs. Metformin is known to inhibit complex I of the respiratory chain [14]. Consequently, reduced ATP (adenosine triphosphate) production leads to the activation of AMPK (adenosine monophosphate kinase), which leads to the downstream inhibition of the mammalian Target of Rapamycin (mTOR), an important activator of malignancy in GB [19].

As expected, treatment with metformin led to a dose-dependent activation of AMPK and inhibition of STAT3 and mTOR in our GB model, as exemplarily shown for TC-11 (Appendix A) and previously demonstrated [20].

Next, we evaluated the effects of increasing doses of metformin on cell death. In our model, 1 µM staurosporine reliably induced apoptosis, as indicated by increased levels of cleaved effector caspase 3 in BTICs and TCs (Appendix A). In contrast, we did not observe a significant increase in caspase cleavage after treatment with increasing doses of metformin, indicating that metformin does not induce caspase-dependent apoptosis. Results from Western Blot analyses were validated using annexin V/7-AAD FACS-analyses. Thereby, the use of the caspase-inhibitor ZVAD (carbobenzoxy-valyl-alanyl-aspartyl-[O-methyl]-fluoromethylketone) reduced staurosporin-induced, but not metformin-induced cell death (Appendix A).

We also investigated autophagy, which is closely related to mTOR signaling [21]. Enhanced conversion of the cytosolic form of microtubule-associated protein 1A/1B-light chain 3 (LC3-I) to LC3-phosphatidylethanolamine conjugate (LC3-II), which is recruited to autophagosomal membranes, is indicative of an induction of autophagy under treatment with metformin [22]. Induction of autophagy was seen in BTIC-11 and TC-18 and to a lesser degree also in TC-11 and was thereby not restricted to progenitor states or a distinct molecular subtype (Appendix A). To confirm results from Western Blot analyses, we used a vector containing the human LC3B gene fused to green fluorescent protein (GFP) to visualize autophagosome formation exemplarily in TC-11. Thereby, a time- and dose-dependent induction of autophagy after treatment with metformin could be confirmed (Appendix A). A significant LC3B-GFP signal even in the medium control may be explained by basal autophagy most likely due to low glucose conditions in cell culture medium.

### 2.5. Metformin Treatment in Mice

To better understand the effects of metformin in vivo, we performed a pilot trial with BTIC-18 that was responsive to metformin in vitro [7]. Orthotopic implantation of BTIC-18 in mice led to a reliable formation of brain tumors that still expressed the proneural progenitor marker SOX-2 (sex determining region Y-box2) [9] and the astrocytic marker GFAP (Figure 5A). Mice were treated with 2.5 or 5 g/L metformin in their drinking water. At 5 g/L metformin, all mice died within a short treatment period. As controls were vital and without tumor-associated symptoms at this time, we assumed that the toxicity of high-dose metformin and not tumor growth led to the death of mice. At 2.5 g/L, metformin treatment was well tolerated, but had no significant effect on tumor growth or size in comparison to controls, and both groups were terminated after 175 days (Figure 5B, logrank test: not significant). In preliminary experiments, metformin was detected and quantified in tissue slices of one treated mouse using tissue extraction with 80% methanol and analysis by HILC-ESI-MS/MS (not shown).

## 3. Discussion

In this work, we examined the general pattern of amino acid metabolism in mesenchymal and proneural BTICs of GB and specifically studied the effects of metformin on amino acid metabolism. We thereby explored extracellular amino acid concentrations of different BTICs, flux of glucose-derived carbon atoms, and effects of metformin on apoptosis and autophagy and further challenged our results in an orthotopic BTIC model.

We found an increased activity of the serine synthesis pathway in proneural BTICs, whereas the investigated mesenchymal BTICs showed an increased flux of glucose-derived carbon atoms into aspartate and glutamate. In contrast to the production of aspartate and glutamate in mesenchymal BTICs, proneural BTICs and TCs took up aspartate and glutamate to fuel the TCA. Treatment with metformin led to a strong reduction in carbon flux from glucose to amino acids and to autophagy, but did not inhibit tumor growth in mice (Appendix A). Metformin might therefore modulate a more protective rather than cell-death inducing mechanism, a finding that fits well with the published literature [23,24].

In previous work, we showed that proneural BTICs metabolize glucose more towards the pentose phosphate pathway [7]. The pentose phosphate pathway is closely linked to the serine cycle [25], and serine metabolism builds the bridge between glycolysis and one-carbon metabolism through upregulation of specific enzymes leading to the generation of serine and antioxidant machinery intermediates. The one-carbon metabolism also comprises a series of interlinking metabolic pathways that generate methyl groups for the synthesis of DNA, polyamines, amino acids, creatine, and phospholipids. Serine hydroxymethyltransferase catalyzes the generation of glycine from serine with the concurrent generation of 5,10-methylenetetrahydrofolate from tetrahydrofolate (THF). Glycine dehydrogenase, also known as glycine decarboxylase, was found most enriched in proneural BTICs in our hallmark analysis. It is the key enzyme of the glycine cleavage system, which catalyzes the degradation of glycine into one-carbon units [17]. Glycine was found to play a key role in rapid cancer cell proliferation in metabolite profiling studies [26]. In glioma, mTORC1 activity was found to regulate post-translational modifications of glycine decarboxylase to modulate glycine metabolism and tumorigenesis [17]. Therefore, enrichment of glycine dehydrogenase in proneural BTICs might lead to the enhanced proliferation of GB.

Members of the family of aldehyde dehydrogenase 1 are closely related to glycolysis [6]. Interestingly, we and others [6,27] found ALD1A3 to be enriched in mesenchymal cells, which was also observed in databases such as the TCGA (Cancer Genome Atlas) and CCGA (Chinese Glioma Genome Atlas) [27]. Members of the aldehyde dehydrogenase (ALDH) superfamily serve as detoxification enzymes as they convert aldehydes to carboxylic acids (reviewed in [18]). ALDH1A3 has also been found to promote the proliferation of glioma stem cells and to regulate the expression of the survival factor tissue transglutaminase in mesenchymal glioma stem cells [28]. An enrichment of aldehyde dehydrogenase (ALDH) in the mesenchymal subgroup led to an increase in glycolytic metabolites [29] and correlated with survival rates of patients with GB [27]. ALD1A3 could therefore be an important modulator of patient survival in mesenchymal BTIC-dominant tumors.

In our stable isotope tracing analyses, we chose low-glucose conditions to mimic the actual situation within GB, where limited glucose is available [30]. Interestingly, BTIC-18 and BTIC-8 consumed all aspartate and glutamate present in cell culture media. In contrast, mean enrichment analyses suggested that mesenchymal BTIC-13 and BTIC-11 metabolized glucose more towards aspartate and glutamate. TC-11 even exported glutamate, a mechanism, which might increase cancer cell dependency on glucose via the glutamate/cystine antiporter SLC7A11 [31]. By exchanging intracellular glutamate for cystine (which is intracellularly reduced to cysteine), SLC7A11 mediates abundancy of the rate-limiting amino acid for glutathione biosynthesis [32]. C-myc induces expression of the SLC7A11 gene, which is upregulated in several cancer types [33] and is a known driver of tumor cell malignancy in glioma [34].

Aspartic acid plays important roles in the biosynthesis of proteins, purines, and pyrimidines, as well as in the urea cycle and as a carrier of reducing equivalents for the electron transport chain in the aspartate-malate shuttle. Glutamic acid, on the other hand, is incorporated into proteins and glutathione, supplies nitrogen for transamination reactions, is secreted from the cell in exchange for other nutrients such as cystine, which, once inside in the cell, is reduced to cysteine, the rate-limiting amino acid for glutathione biosynthesis, and is converted into α-ketoglutarate for tricarboxylic acid cycle anaplerosis [35]. The gene expression profile of lower-grade astrocytomas (mainly of those harboring the IDH1 mutation) indicated that tumor cells might be sensitized to oxidative stress due to reduced glutathione synthesis [36].

Based on previous studies from us and others [7,20,24,37,38,39,40], we chose metformin to further explore GB cell metabolism. Besides its antidiabetic properties, metformin was proposed as a possible antineoplastic agent [41]. Metformin specifically inhibits complex I of the respiratory chain [14], which was also demonstrated in glioma cells [7]. Reduced ATP production leads to the activation of AMPK and inhibition of mTOR as well as several functional inhibitory effects on GB. As those functional effects and their interaction with glucose metabolism have been extensively studied [7], we here chose to focus on amino acid metabolism after treatment with metformin and to link those results to autophagy and apoptosis. Interestingly, treatment with metformin led to a dose-dependent inhibition of glucose-derived carbon flux in nearly all of the investigated lines. Whereas some cells already responded to doses as low as 0.01 mM metformin tid, most cells showed reduced flux starting at 1 mM metformin, and a strong inhibition at a dose of 10 mM metformin. However, a dose of 10 mM metformin is by far higher than the doses reached in humans with standard dose metformin [42] and should therefore be seen more as a positive control rather than a realistic model of the situation in humans. Interestingly, even after 10 mM metformin, glucose-derived ^13^C enrichment in serine and glycine was only moderately reduced in proneural BTIC-18, but strongly reduced in mesenchymal BTIC-11 and BTIC-13, underscoring the dependence of the proneural line on serine and glycine metabolism.

In our study, the lack of caspase 3 activation demonstrated that metformin did not initiate caspase-dependent cell death in proneural and mesenchymal BTICs and TCs However, metformin increased the levels of the autophagic marker LC3 II in subset of cell lines indicating the formation of autophagosomes.

Sesen et al. (2015) also reported an increase in LC3 II and Beclin-1 expression after metformin treatment, indicating the induction of autophagy [13], which is a common phenomenon after inhibition of mTOR [43]. Interestingly, we also found increased annexin V staining after treatment with high-dose metformin, which was unchanged after co-treatment with the caspase inhibitor ZVAD thereby again not indicating caspase-dependent apoptosis.

Autophagy has a dual role in tumorigenesis, depending on the stage of tumor development. Whereas it may serve as a tumor suppressor via degradation of potentially oncogenic molecules in early tumor development, it may also promote the survival of tumor cells by ameliorating stress in the microenvironment in advanced tumor stages [44]. Autophagy is closely related to various other cellular signaling pathways, including amino acid metabolism [45], and amino acids serve as main inhibitors of autophagy [45]. E.g., serine metabolism is closely linked to autophagy by mTOR signaling [46]. Interestingly, BTIC-18, a proneural line with increased serine and glycine metabolism, showed no significant activation of autophagy, even after high-dose metformin.

We found high doses of metformin to be toxic and low doses to be inefficient in vivo, a result that differs from published studies using metformin concentrations of 1–5 g/L [47,48]. We could also not find effects in lower concentrations of metformin in our model, even though other authors have reported significant effects of metformin in orthotopic brain tumor models [39,49] and flank models [50,51]. However, most of them used metformin pre-treated cells [39,50] and i.p. injections [39,49,50,51] of the drug, whereas we used an oral application to mimic the use of metformin in humans. Furthermore, we used metformin in immunocompromised mice and other authors have found that metformin was more active in immunocompetent mice [52].

Our study has several limitations. Our observations on amino acid metabolism of proneural versus mesenchymal BTICs are based on a limited number of heterogeneous lines, which makes general conclusions on subtype-specific metabolic preferences difficult. Therefore, our results should be verified in an independent data set including more lines. Nevertheless, our microarray analyses were based on 36 BTICs and confirmed differences in metabolic hallmarks found in our in vitro assays, including strong enrichment of glycine dehydrogenase among proneural BTICs. We also explored progenitor and differentiated tumor cell states in some assays, which did not, however, show clear trends for amino acid metabolism. Other authors have found a stronger inhibition of tumor cell proliferation in progenitor cells, but not differentiated tumor cells [40]; however data on amino acid metabolism and autophagy on this aspect are lacking completely. Even though we explored several BTICs of the proneural and mesenchymal subtype, we were not able to perform the same assays with BTICs of the classical subtype as we were lacking primary lines of the classical subtype. However, other authors have observed that metabolism of proneural and mesenchymal BTICs was most distinct [6]. Further studies should explore, if the preferential activity of amino acid profiles of proneural and mesenchymal BTICs are maintained when both BTIC types are present. In glioblastoma, cells of distinct molecular subtypes may be present within one tumor, but with regional preferences. For example, previous studies showed that invasive glioblastoma cells carry a predominantly mesenchymal genotype, whereas the other subtypes were more present in the tumor bulk [53].

Our study also has several strengths. Whereas a preference of mesenchymal cells for glycolysis has already been described in studies including numerous BTIC cultures [6,7,11], our study is the first one to demonstrate increased serine and glycine metabolism among proneural BTICs along with an increased activity of the pentose phosphate pathway [7]. Our results were based on several in-depth metabolic assays and related to microarray data from multiple lines and published studies.

## 4. Materials and Methods

### 4.1. Cell Based Methods

#### 4.1.1. Tumor Specimens and Enrichment of BTICs

We used primary BTICs that were isolated from freshly resected human gliomas, as described (10). Specimen sampling and BTIC culture were approved by the Ethics Committee of the University of Regensburg (No° 09/101) and all patients gave written informed consent. BTICs were kept in DMEM low glucose medium (DMEM with 1 g/L of glucose; Sigma-Aldrich, St. Louis, MO, USA, #D6046) containing Epidermal Growth Factor (Miltenyi Biotec, Bergisch Gladbach, Germany, #130-097-751) and Fibroblast Growth Factor (Miltenyi Biotec, Bergisch Gladbach, Germany, #130-093-842) supplemented with 50 U (*v*/*v*) Penicillin, 0.05% (*v*/*v*) Streptomycin (#P4333), 2 mM (v/v) L-Glutamine (#G7513), 1% (*v*/*v*) MEM Vitamin Solution (#M6895), and 1% (*v*/*v*) non-essential amino acids (#M7145) (all Sigma-Aldrich, St. Louis, MO, USA). For differentiation, growth factors were withdrawn, and cells were exposed to 10% fetal calf serum (FCS) for at least two weeks. Cells were incubated at 37 °C, 5% CO_2_, 95% humidity in a standard tissue culture incubator.

#### 4.1.2. Treatment of BTICs with Metformin

Metformin hydrochloride (Sigma-Aldrich, St. Louis, MO, USA, #PHR1084) was dissolved in cell culture medium and used at indicated doses.

#### 4.1.3. Annexin-V/7-AAD FACS Analysis

After treatment, cells were stained with Annexin-V-FITC and 7-amino-actinomycin D (7-AAD) (BioLegend, San Diego, CA, USA) according to the manufacturer’s instructions [54]. The cell-permeant pan-caspase inhibitor ZVAD-FMK was used to explore effects of apoptosis inhibition after different treatments [55].

Flow cytometric analyses were performed on a FACSCanto II (BD Biosciences, Franklin Lakes, NJ, USA) using BD CellQuestPro for data acquisition and analysis. Final processing and analysis were performed with FlowJo v9.5.3 software.

#### 4.1.4. LC-3 I/II Staining

To visualize autophagosome formation in real time in live cells, we established an LC3-GFP-reporter assay (pSELECT-GFP-hLC3, InvivoGen, Toulouse, France, #psetz-gfplc3, 1.076 µg/µL). Autophagosome formation was monitored over 48 h visualized using fluorescence microscopy.

### 4.2. Protein-Based Methods

#### Western Blot

To investigate the protein levels of (p)mTOR (mammalian target of rapamycin), (p)AMPK (adenosine-monophosphate kinase), (p)STAT3 (signal transducer and activator of transcription 3), LC3 I/II, Caspase 3, cleaved Caspase 3, and GAPDH (glyceraldehyde 3-phosphate dehydrogenase), whole-cell lysates were prepared with RIPA buffer and 30 µg of each lysate was subjected to Western blotting on a denaturing 10% SDS-PAGE. A total of 30 µg of total cell lysate were diluted in Laemmli buffer, separated on a 10% SDS-PAGE gel, and transferred to nitrocellulose membranes by semi-dry blotting. The membranes were blocked with 5% milk powder or 5% BSA in 0.02% Tween TBS (TBST) for 1 h. Membranes were incubated with specific monoclonal antibodies for STAT3 (1:1000, #9145), pSTAT3 (1:1000, phosphorylation site Y705, #9132), LC3 I/II (1:1000, #NB100-2220, Novus Biologicals, Centennial, CO, USA), cleaved Caspase 3 (1:1000, #9664), mTOR (1:1000, #2983), pmTOR (1:1000, #5536), AMPKα (1:1000, #2603), pAMPKα (1:1000, #2535 all Cell Signaling, Cambridge, UK) or GAPDH (1:2000, #sc-48167 from Santa-Cruz, Dallas, TX, USA) overnight at 4 °C. Immunocomplexes were visualized using horseradish peroxidase-conjugated antibodies, mouse anti-rabbit IgG-HRP (#sc-2357 Santa-Cruz, Dallas, TX, USA), goat anti-rabbit (Advansta, San Jose, CA, USA #R-05072-500) or donkey anti-goat (Santa-Cruz, Dallas, TX, USA, #sc-2020) followed by enhanced chemoluminescence (Western HRP Substrate, #WBLUFO100, Millipore, Darmstadt, Germany).

### 4.3. mRNA-Based Methods

#### Microarray Analysis, Clustering, and Gene Set Enrichment Analysis

Microarray analysis was performed as described elsewhere [56]. Computational analysis was performed using R and Bioconductor (http://www.bioconductor.org (accessed on 11 October 2015)). Genes differentially expressed between mesenchymal and proneural BTICs were identified using the package Limma [56]. We computed a ranked gene list based on logarithmic fold change expression and applied the GSEAPreranked tool of GSEA v2.2.2 (www.broadinstitute.org/gsea, accessed on 1 December 2015). Gene set collections C2, C5, and C6 from the MSigDB (www.broadinstitute.org/msigdb, accessed on 1 December 2015) were used within GSEA. Enriched gene sets with an absolute normalized enrichment score >2 and a maximum FDR < 0.25% were considered for further evaluation. Data are saved in the gene expression omnibus (GEO) functional genomics data repository under the accession numbers GSE51305 and GSE76990.

### 4.4. Metabolic Methods

#### 4.4.1. Determination of Extracellular Metabolite Concentrations

For measurement of extracellular metabolites, a 10-µL aliquot of the cell culture supernatant was used for amino acid analysis by HPLC-ESI-MS/MS (AB SCIEX, Darmstadt, Germany) after propyl chloroformate/propanol derivatization as described [57].

Quantification was achieved based on multi-point calibration curves for each analyte using the corresponding stable isotope labeled analog as internal standard.

#### 4.4.2. ^13^C-Glucose Isotope Tracing

For stable isotope tracing experiments, cells were harvested, and extracts prepared. Therefore, samples were separated into extract and sample residues by centrifugation at 9560× *g*. All extracts were combined before evaporation to complete dryness. The sample residue was reconstituted in 100 μL water. A 10-µL aliquot was used for amino acid analysis by HPLC-ESI-MS/MS (AB SCIEX, Darmstadt, Germany) after derivatization using propyl chloroformate/propanol. We used [^13^C_6_]glucose as a tracer substrate and measured metabolites and amino acids derived therefrom. For amino acid analysis, one transition for each possible isotopologue was measured. From the GC-MS full scan data, extracted ion chromatograms based on the m/z values of the individual isotopologues were used for data analysis. Stable isotope tracing data were corrected for natural abundance of ^13^C and tracer impurity using IsoCorrectoR [58].

### 4.5. Orthotopic BTIC Mouse Model

#### 4.5.1. Preparation and Conduct of Mouse Model

Two groups of NOD.Cg-Prkdcscid Il2rgtm1Wjl/SzJ mice were anesthetized before all intracranial procedures and placed in a stereotaxic fixation device (Stoelting, Wood Dale, IL, USA). A burr hole of 0.5 mm was drilled in the skull 1 mm lateral and 2 mm to the front of the bregma. The needle of a Hamilton syringe (Hamilton, Darmstadt, Germany) was introduced to a depth of 3 mm. A total of 1 × 10^5^ BTIC resuspended in 2 μL phosphate-buffered saline (PBS) were slowly injected into the brain. In the experimental groups, mice were treated with either 5 or 2.5 g/L metformin in drinking water starting on day 5 after the operation. The mice were observed and weighed daily and euthanized upon development of neurological symptoms or progressive weight loss greater than 20%.

The experiments were performed according to the German Animal Protection law and registered under the number AZ 54-2532.1-28/14. Overall survival (OS) was determined in days from the date of tumor cell implantation. Kaplan–Meier curves were plotted to estimate OS as a function of treatment. Group differences were assessed by the log-rank method.

#### 4.5.2. H&E Staining and Immunofluorescence Staining for MCT4, GFAP and SOX-2

Mice brains were collected, paraffin-embedded, and sliced. Adjacent slices were stained with H&E or immunostained using MCT4 (1:100, #sc-376465, Santa-Cruz, Dallas, TX, USA), GFAP (1:500, #Z0334, DAKO, Hamburg, Germany) and SOX-2 (1:100, #sc-376465, Santa-Cruz, Dallas, TX, USA) specific antibodies (secondary antibodies: donkey anti-goat + AF488, 1:400, #A-32814, Life Technologies, Carlsbad, CA, USA; donkey anti-mouse + AF488, 1:400, #A-21202, Life Technologies, Carlsbad, CA, USA; donkey anti-rabbit IgG + AlexaFluor, 1:400, #A-21206, Life Technologies, Carlsbad, CA, USA).

### 4.6. Statistics

All statistical analyses were performed using GraphPad Prism Version 7.05 (GraphPad Software, Inc. San Diego, CA, USA) if not otherwise indicated.

We calculated a two-way ANOVA to compare the results (mean values and SDs) of controls and treated BTICs or between groups of BTICs and TCs or the corresponding BTIC/TC pair. All assays were performed in triplicate. We used Tukey’s post-hoc test to control for multiple comparisons. The level of significance was set at *p* < 0.05 (asterisks indicate * *p* < 0.05, ***p* < 0.01, *** *p* < 0.001, and **** *p* < 0.0001). Western blots were repeated three times with at least two biological replicates and quantified using Image J, version 1.49.

## 5. Conclusions

We observed that proneural BTICs had increased activity of the serine and glycine pathway, whereas the mesenchymal BTICs in our study utilized glucose preferably for the synthesis of aspartate and glutamate. Metformin treatment led to increased autophagy and strong inhibition of carbon flux from glycolysis to amino acids. However, treatment with metformin was toxic at high doses in vivo and inefficient at lower doses in an immunocompromised mouse model. Further studies are warranted to better understand the multilayer mode-of-action of metformin and possible resistance mechanisms before metformin might be proposed as repurposed agent in the treatment of glioma.

## Figures and Tables

**Figure 1 ijms-24-03199-f001:**
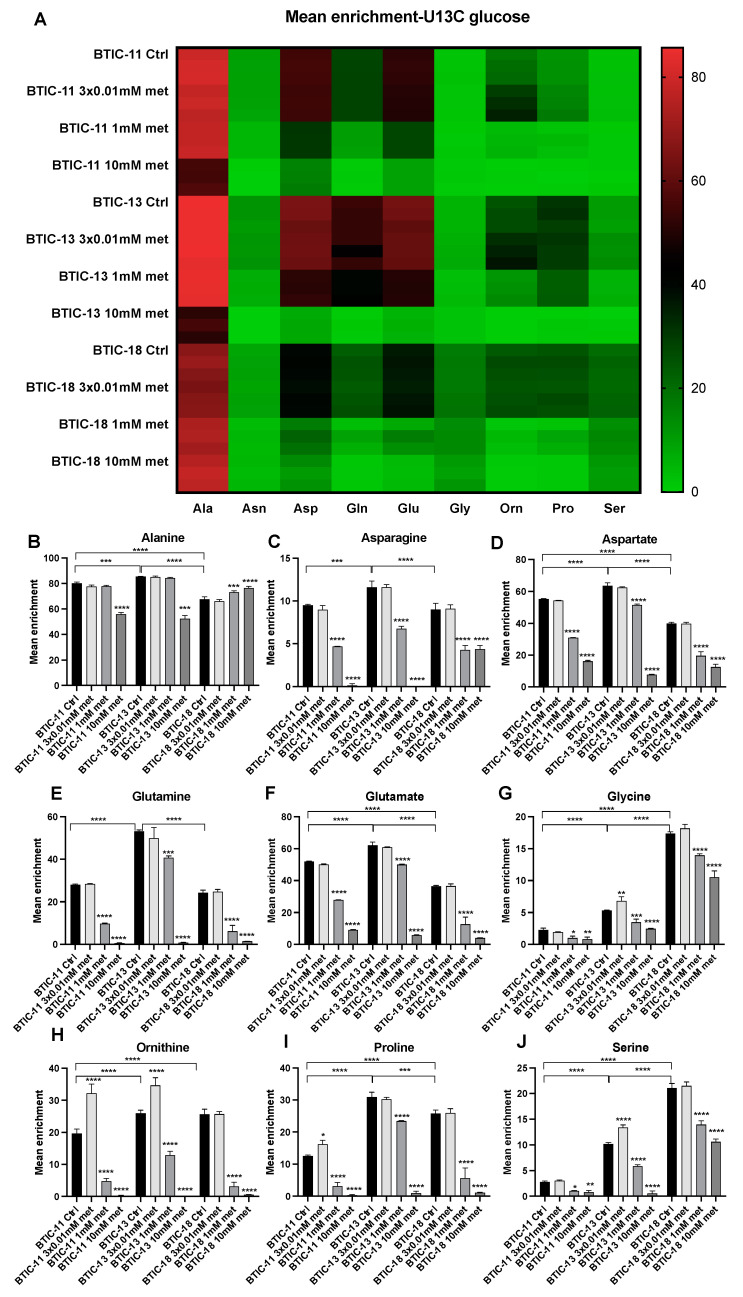
Stable isotope tracing using [^13^C6]glucose. Mean enrichment of isotope-labeled amino acids was determined by HPLC-MS/MS. The heatmap allows a comparison between the different amino acids (**A**) in BTIC-11, BTIC-13 and BTIC-18. Graphs for single amino acids allow a valid comparison between the different lines and treatments with metformin. Thereby, mean enrichments of alanine (**B**), asparagine (**C**), aspartate (**D**), glutamine (**E**), glutamate (**F**), glycine (**G**), ornithine (**H**), proline (**I**) and serine (**J**) are depicted for mesenchymal BTIC-11 and BTIC-13, and proneural BTIC-18. All assays were performed in triplicate. Brackets with asterisks indicate the comparison of different BTICs. Asterisks directly above the bar indicate significance of metformin treated cells versus their respective control. Asterisks indicate * *p* < 0.05, ** *p* < 0.01, *** *p* < 0.001, and **** *p* < 0.0001.

**Figure 2 ijms-24-03199-f002:**
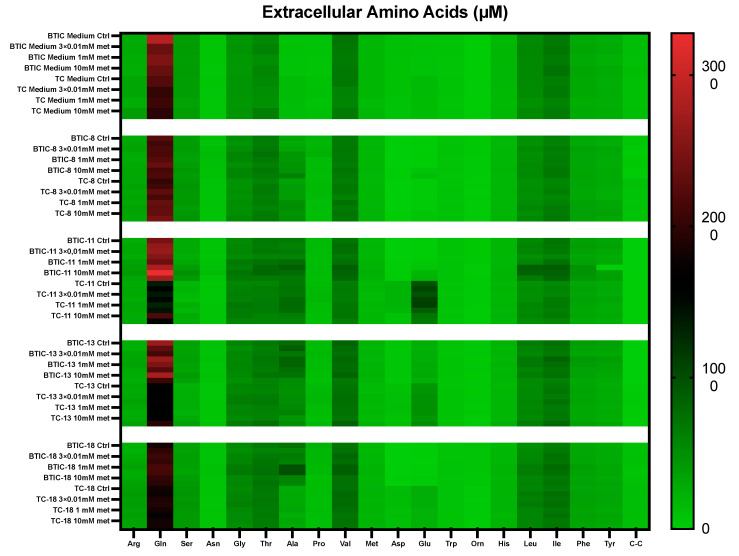
Extracellular amino acids. Extracellular amino acids were measured by HPLC-ESI-MS/MS after 48 h. Cells were either left untreated or treated with increasing doses of metformin as indicated. The heatmap shows concentrations of all indicated amino acids in cell culture medium for proneural BTIC- and TC-8 and -18, and mesenchymal BTIC- and TC-11 and -13. Arg = arginine, Gln = glutamine, Ser = serine, Asn = asparagine, Gly = glycine, Thr = threonine, Ala = alanine, Pro = proline, Val = valine, Met = methionine, Asp = aspartate, Glu = glutamate, Trp = tryptophan, Orn = ornithine, His = histidine, Leu = leucine, Ile = isoleucine, Phe = phenylalanine, Tyr = tyrosine, C-C = cysteine. All assays were performed in triplicate.

**Figure 3 ijms-24-03199-f003:**
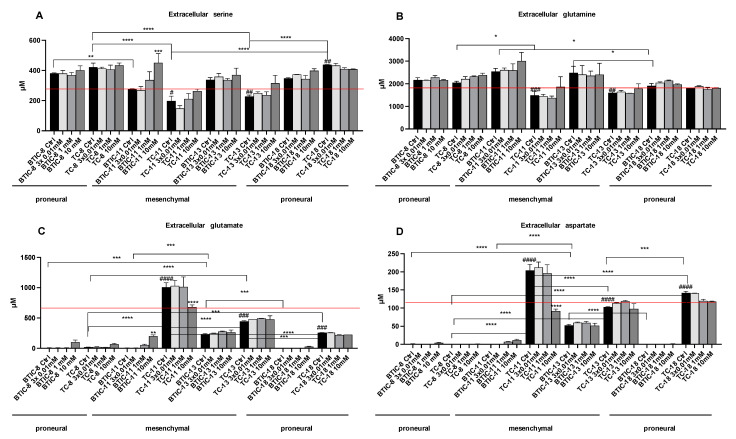
Bar graphs show abundance of (**A**) serine, (**B**) glutamine, (**C**) glutamate, and (**D**) aspartate in cell culture supernatants. Extracellular amino acids were measured by HPLC-ESI-MS/MS after 48 h. Cells were either left untreated or treated with increasing doses of metformin as indicated. Red lines indicate amino acid concentrations in cell culture medium without cells. All assays were performed in triplicate. Significant results are only depicted for valid comparisons. Brackets with asterisks indicate the comparison of different BTICs or the comparison of different TCs. Asterisks directly above the bar indicate significance of metformin treated cells versus their respective controls. Pound signs indicate significance when comparing the corresponding BTIC and TC pair. Asterisks indicate * *p* < 0.05, ** *p* < 0.01, *** *p* < 0.001, and **** *p* < 0.0001. Pounds indicate # *p* < 0.05, ## *p* < 0.01, ### *p* < 0.001, and #### *p* < 0.0001.

**Figure 4 ijms-24-03199-f004:**
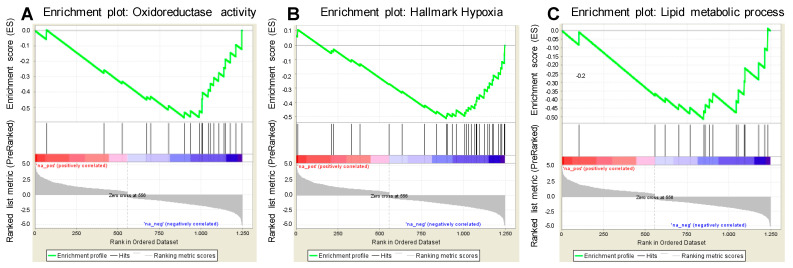
Enrichment plots of the hallmarks (**A**) hypoxia, (**B**) lipid metabolic process, and (**C**) oxidoreductase activity to compare gene expression of proneural and mesenchymal BTICs. The green line in the upper panel indicates the enrichment profile, whereas rank metrics are depicted in the lower part of the graphs.

**Figure 5 ijms-24-03199-f005:**
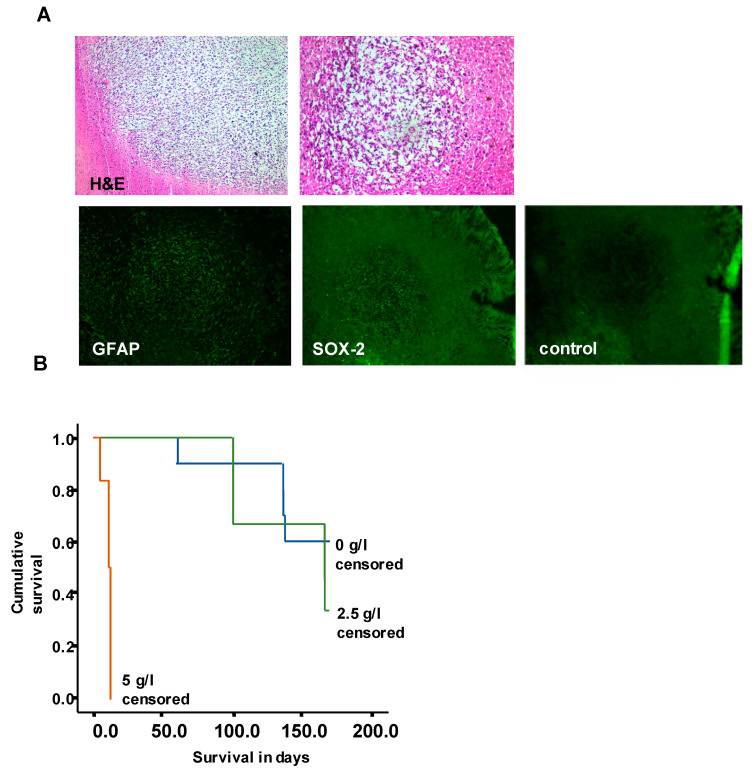
(**A**) Orthotopic implantation of BTIC-18 led to the formation of GFAP and SOX-2 positive tumors in mice. Immunofluorescence staining for GFAP and SOX was accompanied by an IgG-negative control. (**B**) Oral treatment with metformin at a dose of 5 g/L metformin in drinking water was toxic in a mouse model of NOD-SCID mice, whereas 2.5 g/L metformin did not prolong the survival of mice bearing orthotopic gliomas derived from BTIC-18. *n* = 10 for controls, *n* = 9 for metformin treatment.

## Data Availability

Microarray data are deposited at the gene expression omnibus (GEO) functional genomics data repository under the accession numbers GSE51305 and GSE76990. Further details and other data that support the findings of this study are available from the corresponding author upon request.

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
