# Peer review of "Heterogeneity of Amino Acid Profiles of Proneural and Mesenchymal Brain-Tumor Initiating Cells"

_ijms, 2023, doi:10.3390/ijms24043199_

Round 1

Reviewer 1 Report

The ijms-2169466 Heterogeneity of amino acid profiles of proneural and mesenchymal brain-tumor initiating cells” and Authors:Corinna Seliger-Behme , Lisa Rauer , Anne-Louise Wüster , Silvia Moeckel , Verena Leidgens , Birgit Jachnik , Laura-Marie Ammer , Katja Dettmer , Markus J. Riemenschneider , Peter J. Oefner , Martin Proescholdt , Arabel Vollmann-Zwerenz , Peter Hau. This manuscript reports the study of heterogeneity of amino acid profiles of proneural and mesenchymal brain-tumor initiating cells. The manuscript could be of interest for publication in "International Journal of Molecular Sciences". The article focuses mainly on the investigated basic amino acid profiles of proneural and mesenchymal BTICs to explore a potential distinct utilization and biosynthesis in these subgroups, and measured extracellular amino acid concentrations of different BTICs at baseline and after treatment with metformin. It can be considered for publication.

1.  Abstract: Add one sentence of conclusion at the end of the paragraph.

2.  Keywords Should be:glioma; metabolism; metformin; proneural and mesenchymal brain; tumor initiating cells 

3. Line 104, 157, 474:*p<0.05, **p<0.01, ***p<0.001, and ****p<0.0001. should *p<0.05, **p<0.01, ***p<0.001, and ****p<0.0001.

4.  Line 380-383: add the reference.

Author Response

(…) This manuscript reports the study of heterogeneity of amino acid profiles of proneural and mesenchymal brain-tumor initiating cells. The manuscript could be of interest for publication in "International Journal of Molecular Sciences". The article focuses mainly on the investigated basic amino acid profiles of proneural and mesenchymal BTICs to explore a potential distinct utilization and biosynthesis in these subgroups, and measured extracellular amino acid concentrations of different BTICs at baseline and after treatment with metformin. It can be considered for publication.

Author’s response: Thank you for your excellent review and overall positive evaluation of our manuscript. Please find below our responses to the points raised during the review process.

  1. Abstract: Add one sentence of conclusion at the end of the paragraph.

Author’s response: We added a conclusion at the end of the paragraph as follows: “In conclusion, we found distinct amino acid profiles of proneural and mesenchymal BTICs, and inhibitory effects of metformin on BTICs in vitro. However, further studies are warranted to better understand potential resistance mechanisms against metformin in vivo.”

  1. Keywords Should be: glioma; metabolism; metformin; proneural and mesenchymal brain; tumor initiating cells 

Author’s response: We changed the keywords as suggested.

  1. Line 104, 157, 474: *p<0.05, **p<0.01, ***p<0.001, and ****p<0.0001. should *p<0.05, **p<0.01, ***p<0.001, and ****p<0.0001.

Author’s response: We now indicated the “p” in italics.

  1. Line 380-383: add the reference.

Author’s response: We added two references as suggested. 

Reviewer 2 Report

Here C. Seliger et al examined the amino acid (AA) metabolism profile of brain-tumor-initiating cells (BTICs) following Metformin treatment. Besides identifying the difference in AA metabolism signature of proneural or mesenchymal BTICs, the authors linked it to increased autophagy and inhibition of carbon flux from glucose. The authors further tested oral administration of metformin to mice orthotopic model of GB. 

Overall, while the authors presented quite a lot of results, their presentation and description are relatively vague. It is difficult to understand the results presented herein. I would suggest the acceptance of the article only following sufficient revision below:

- The authors showed preferential activity of AA metabolism between proneural or mesenchymal BTICs. Considering the heterogenous nature of GB, however, have the authors tested whether such preferential activity is masked (or lost) when both BTIC types are present?

- In page 7 line 208-209, the authors explained result regarding LC3B gene-fused with GFP to visualize authophagosome. It is not mentioned which results it is referring to? Fig S3A? Please improve the resolution of the fluorescence image. Moreover, it is unclear why there is significant LC3B-GFP signal even in medium control group. 

- The authors presented the orthotopic GB mice model and results very vaguely. How long was the incubation period between BTIC injection and metformin treatment? Have the authors confirmed that Metformin indeed reach and sufficiently accumulate to the GB tumor? Was there any change in tumor size, or at least change in apoptotic rate & molecular phenotype? 

Author Response

Here C. Seliger et al examined the amino acid (AA) metabolism profile of brain-tumor-initiating cells (BTICs) following Metformin treatment. Besides identifying the difference in AA metabolism signature of proneural or mesenchymal BTICs, the authors linked it to increased autophagy and inhibition of carbon flux from glucose. The authors further tested oral administration of metformin to mice orthotopic model of GB. 

Overall, while the authors presented quite a lot of results, their presentation and description are relatively vague. It is difficult to understand the results presented herein. I would suggest the acceptance of the article only following sufficient revision below:

Author’s response: Thank you for your thorough evaluation of our manuscript. We changed all points as suggested.

- The authors showed preferential activity of AA metabolism between proneural or mesenchymal BTICs. Considering the heterogenous nature of GB, however, have the authors tested whether such preferential activity is masked (or lost) when both BTIC types are present?

Author’s response: Our study was based on several BTICs of proneural and mesenchymal origin derived from fresh tumor resections. Thereby, the tested lines are derived from several glioblastomas with inter- and intratumoral heterogeneity. We did not co-culture proneural and mesenchymal BTICs after BTIC isolation to further explore the maintenance of amino acid profiles after co-cultivation. We added this point to our discussion section as follows: “Further studies should explore, if the preferential activity of amino acid profiles of proneural and mesenchymal BTICs are maintained when both BTIC types are present. In glioblastoma, cells of distinct molecular subtypes may be present within one tumor, but with regional preferences. For example, previous studies showed that invasive glioblastoma cells carry a predominantly mesenchymal genotype, whereas the other subtypes were more present in the tumor bulk.”

- In page 7 line 208-209, the authors explained result regarding LC3B gene-fused with GFP to visualize authophagosome. It is not mentioned which results it is referring to? Fig S3A? Please improve the resolution of the fluorescence image. Moreover, it is unclear why there is significant LC3B-GFP signal even in medium control group. 

Author’s response: Thank you for mentioning this important point. The results regarding LC3B gene-fused with GFP to visualize authophagosome formation refer to Supplementary Figure 3A. We added this information to the results part. We improved the resolution of the fluorescence image. A significant LC3B-GFP signal even in the medium control may be explained by basal autophagy most likely due to low glucose conditions in cell culture medium (page 7, line 211).

- The authors presented the orthotopic GB mice model and results very vaguely. How long was the incubation period between BTIC injection and metformin treatment? Have the authors confirmed that Metformin indeed reach and sufficiently accumulate to the GB tumor? Was there any change in tumor size, or at least change in apoptotic rate & molecular phenotype?

Author’s response: The incubation period between BTIC injection and metformin treatment was 5 days. In preliminary experiments, we were able to measure metformin in the brain. We therefore added to page 7, lines 227-229: “In preliminary experiments, metformin was detected and quantified in tissue slices of one treated mouse using tissue extraction with 80% methanol and analysis by HILC-ESI-MS/MS (not shown).“ These experiments are currently extended to a larger sample size and to amino acid measurements and will be published independently. In line with survival data, there was also no significant change in tumor size among metformin-treated and untreated mice (“At 2.5 g/L, metformin treatment was well tolerated, but had no significant effect on tumor growth or size in comparison to controls” page 7, line 220). The molecular phenotype was constant as shown for example for SOX-2 expression in proneural BTICs (page 7, line 215). We did not measure apoptotic rate in the orthotopic GB model, but tumor histology (HE, fig. 5A) showed typical GB features with intratumoral necrosis.

Please note that we have also added one new author who has performed the experiments on metformin levels in the mouse brains.

Round 2

Reviewer 2 Report

The authors have sufficiently addressed my comments and have amended the manuscript with these clarifications. I understand that the authors would like to publish separately the result of orthotopic GB mice, and thus can only mention preliminary data. 

I believe the article is of significant quality for IJMS, and I would recommend the acceptance of this article in its current form.